# *MYBL2* is a sub-haploinsufficient tumor suppressor gene in myeloid malignancy

Stefan Heinrichs[1,2]*, Lillian F Conover[2], Carlos E Bueso-Ramos[3], Outi Kilpivaara[4], Kristen Stevenson[5], Donna Neuberg[5], Mignon L Loh[6], Wen-Shu Wu[7], Scott J Rodig[8], Guillermo Garcia-Manero[9], Hagop M Kantarjian[9], A Thomas Look[2]

[1]Institute of Transfusion Medicine, University Hospital Essen, Essen, Germany; [2]Department of Pediatric Oncology, Dana-Farber Cancer Institute, Boston, United States; [3]Department of Hematopathology, MD Anderson Cancer Center, Houston, United States; [4]Department of Medical Genetics, University of Helsinki, Helsinki, Finland; [5]Department of Biostatistics and Computational Biology, Dana-Farber Cancer Institute, Boston, United States; [6]Department of Pediatrics, University of California, San Francisco, San Francisco, United States; [7]Center for Cell Therapies, Children's Hospital Oakland Research Institute, Oakland, United States; [8]Department of Pathology, Brigham and Women's Hospital, Boston, United States; [9]Department of Leukemia, MD Anderson Cancer Center, Houston, United States

*For correspondence: stefan.
heinrichs@uk-essen.de

Competing interests: The authors declare that no competing interests exist.

**Abstract** A common deleted region (CDR) in both myelodysplastic syndromes (MDS) and myeloproliferative neoplasms (MPN) affects the long arm of chromosome 20 and has been predicted to harbor a tumor suppressor gene. Here we show that *MYBL2*, a gene within the 20q CDR, is expressed at sharply reduced levels in CD34+ cells from most MDS cases (65%; n = 26), whether or not they harbor 20q abnormalities. In a murine competitive reconstitution model, *Mybl2* knockdown by RNAi to 20–30% of normal levels in multipotent hematopoietic progenitors resulted in clonal dominance of these 'sub-haploinsufficient' cells, which was reflected in all blood cell lineages. By 6 months post-transplantation, the reconstituted mice had developed a clonal myeloproliferative/ myelodysplastic disorder originating from the cells with aberrantly reduced *Mybl2* expression. We conclude that downregulation of *MYBL2* activity below levels predicted by classical haploinsufficiency underlies the clonal expansion of hematopoietic progenitors in a large fraction of human myeloid malignancies.

## Introduction

The molecular changes underlying human myeloid malignancies remain difficult to unravel, posing major obstacles to the development of effective countermeasures. Although the silencing of tumor suppressor genes by chromosomal deletions, point mutations, or other mechanisms is an acknowledged factor in myeloid cell transformation, the specific involvement of gene dosage is not well understood. In broadest terms, single-copy loss of a suppressor gene can be sufficient to modify gene function and promote tumorigenesis (classical haploinsufficiency), while in other tumors, the loss of two alleles is required (two-hit paradigm of Knudson) (*Knudson, 1971*). Recent evidence indicates that more subtle reductions in suppressor gene function may contribute importantly to myeloid malignancy (*Rosenbauer et al., 2004*; *Liu et al., 2007*), leading to the need for faithful animal models to establish that such changes are truly involved in tumorigenesis.

Loss of an interstitial segment of chromosome 20q (20q CDR) is detected in about 4% of myelodysplastic syndromes (MDS) (*Haase et al., 2007*), and this region is variably affected in different types of myeloproliferative neoplasms (MPN), including polycythemia vera (10%) and primary myelofibrosis

**eLife digest** Blood cells are produced within bone marrow by specialized stem cells and progenitor cells. Abnormalities in this process lead to a group of diseases known as myeloid malignancies, which include acute myeloid leukaemia—in which the bone marrow produces abnormal white blood cells—and myelodysplastic syndromes, which are caused by too few mature blood cells being produced.

Many individuals affected by these disorders possess a shortened form of chromosome 20 that lacks a number of genes. This deletion is only ever seen in one of their two copies of the chromosome—suggesting that at least some of these genes are essential for survival—but the identity of the gene(s) that are associated with the increased risk of myeloid malignancies is unknown.

Now, Heinrichs et al. have uncovered a key tumor suppressor among those genes frequently lost on chromosome 20. The gene, which is called *MYBL2*, encodes a transcription factor that helps to control the cell division cycle. Myeloid malignancy patients lacking one copy of this gene showed levels of *MYBL2* expression that were less than 50% of those in healthy individuals. This suggests that additional mechanisms must be acting to reduce expression of their remaining copy of the gene. Surprisingly, *MYBL2* levels were also reduced in myeloid malignancy patients who possessed two intact copies of chromosome 20, indicating that loss of a single copy represents only one mechanism to reduce *MYBL2* expression, i.e., the 'tip-of-the-iceberg'. Hence, this finding reveals a more general role for *MYBL2* as it indicates that more patients are likely to be affected by altered expression of this gene.

To confirm their findings from studies in patients, Heinrichs et al. used gene silencing techniques to reduce the expression of *MYBL2* in mice and showed that this induced symptoms of myeloid malignancies in the animals. Moreover, injection of modified cells from these animals into healthy mice also induced symptoms in the recipients. The modified cells are able to expand more robustly than normal cells, and this dominance induced by downregulation of the tumor suppressor increases the risk of malignancy.

In addition to revealing a new tumor suppressor gene and its contribution to myeloid malignancies, the study by Heinrichs et al. highlights the importance of gene dosage in mediating the effects of tumor suppressors.

(12%), and less commonly in acute myeloid leukemia (AML; 1%) (*Bench et al., 2000*). Notably, only heterozygous deletions have been found in studies of myeloid malignancies with loss of chromosome 20q, without any evidence of homozygous deletion or mutations of a gene within the affected region (*Heinrichs et al., 2009*; *Huh et al., 2010*). These findings implicate a gene within the 20q CDR that is essential for cell viability, but whose tumor suppressor function is strongly dose-dependent and does not follow the classical Knudson model (*Knudson, 1971*), which predicts biallelic gene inactivation. Instead, monoallelic loss, with or without additional epigenetic or microRNA (miRNA)-mediated downregulation of the remaining allele, may reduce gene expression levels sufficiently to promote myeloid cell transformation. Thus, we sought to identify candidate tumor suppressor genes within the 20q CDR on the basis of their reduced expression in malignant myeloid progenitor cells, as we have reported previously for *CTNNA1* in myeloid malignancies with deletions of chromosome 5q (*Liu et al., 2007*; *Ye et al., 2009*).

Here we identify *MYBL2*, which encodes a transcription factor with functions in checkpoint control of the G2 cell cycle phase, as a key tumor suppressor gene in over one half of all MDS cases, whether characterized by a 20q deletion or a normal karyotype. Reductions of *MYBL2* dosage to levels below those commensurate with single-copy loss conferred a competitive advantage to hematopoietic progenitor cells in both primary and secondary transplantation assays and were associated with histopathologic changes typical of myeloid neoplasia. These findings implicate aberrantly low levels of *MYBL2* expression as a central mechanism in the development of clonal dominance in MDS and other myeloid malignancies.

## Results

### *MYBL2* identified as a potential tumor suppressor

We first studied the gene expression profiles of CD34+ hematopoietic progenitor cells from eight MDS cases with cytogenetically evident aberrations of chromosome 20q, as compared to CD34+ cells

from normal individuals (*Figure 1—figure supplement 1A*). We found that *MYBL2*, which resides near the center of this region (*Figure 1—figure supplement 2*), was among the top-scoring genes downregulated in MDS cells compared to normal CD34+ controls. *MYBL2* encodes a highly conserved transcription factor that acts as a major component of the dREAM complex, which controls the G2-to-M phase transition within the cell cycle (*Korenjak et al., 2004*; *Lewis et al., 2004*). The mean expression level of *MYBL2* (39%) was less than that of normal CD34+ cells (*Figure 1—figure supplement 1B*), suggesting that mechanisms beyond the deletion of one allele might affect the remaining allele. Thus, on the assumption that some MDS cases with a normal karyotype might also have reduced expression of a gene or genes within the 20q CDR, we undertook an expression analysis of CD34+ cells from 18 MDS cases with a normal karyotype and the eight cases with a 20q aberration (*Supplementary file 1A*), compared to CD34+ cells from four normal bone marrow samples. *MYBL2* was the top-scoring gene among 108 differentially expressed genes (*Figure 1A*). Further analysis showed that in 17 (65%) of 26 MDS patients (*Figure 1—figure supplement 3A*), *MYBL2* expression levels in CD34+ cells were reduced to ≤45% of normal CD34+ cells levels (mean 32.8%). These microarray data were confirmed by QRT-PCR analysis of *MYBL2* expression levels in 10 MDS cases and eight normal CD34+ bone marrow samples, including four additional normal control CD34+ samples that were not part of the original microarray analysis (*Figure 1B*). Importantly, *MYBL2* levels measured by QRT-PCR in MDS samples were highly correlated (r = 0.966) with those determined by microarray analysis (*Figure 1—figure supplement 3B*). Hence, *MYBL2* appears to be expressed at reduced levels in nearly two-thirds of MDS cases, including those with a normal karyotype, suggesting that it functions as a dose-dependent tumor suppressor gene.

To address whether reduced levels of *MYBL2* expression actively contribute to the MDS phenotype or merely constitute a compensatory response to other abnormalities, we asked if reduced MYBL2 transcriptional activity is reflected in the MDS gene expression signature. To model gene dose insufficiency by RNA interference (RNAi), we used a set of eight shRNA-expression vectors in the MDS/AML cell line SKM1, which reduced *MYBL2* expression to 5–30% of the endogenous *MYBL2* levels (*Figure 2A*), approximating the endogenous levels we had identified in CD34+ cells from patients (*Figure 1B*). To identify the genes most highly correlated with decreased *MYBL2* expression, we introduced our shRNA-expression vectors into normal primary human CD34+ cells and measured gene expression at 48 hr post-transduction. Nearly all differentially expressed genes were downregulated after *MYBL2* knockdown (81 of 89 genes; *Figure 2B* and *Figure 2—figure supplement 1*). Because these results are based on eight different *MYBL2*-specific shRNAs, each with different knockdown efficiency, this analysis ensures the robustness of the data and eliminates potentially confounding off-target effects by individual shRNAs. Gene set enrichment analysis (GSEA) (*Subramanian et al., 2005*) identified mitosis as the key biological process reflected by the *MYBL2* gene signature, specifically the G2-to-M phase transition (*Figure 2—figure supplement 2*) (*Whitfield et al., 2002*; *Zhu et al., 2004*; *Shepard et al., 2005*). Individual genes and their functions included the G2 cell cycle regulator cyclin B (*CCNB1* and *CCNB2*), the early mitotic regulator *FBXO5* and the coregulator of chromosome architecture *CDCA2* (*Supplementary file 1B*).

We finally performed GSEA to assess whether this *MYBL2* gene expression signature (81 genes, *Supplementary file 1B*) was evident in MDS samples with intrinsically low expression of this gene. This analysis revealed robust enrichment of the *MYBL2* signature in MDS CD34+ cells (*Figure 2C*). Furthermore, the expression levels of *MYBL2*-regulated genes corresponded closely with those of *MYBL2* itself (*Figure 2D*). Together, these findings indicate that *MYBL2* downregulation in CD34+ cells is reflected in the aberrant gene expression profile of a large fraction of MDS cases.

## Low *MYBL2* expression in cases with a normal karyotype

To determine which MDS patients with a normal karyotype have an '*MYBL2*-low' expression signature, we first analyzed the expression levels of MYBL2-regulated genes in the CD34+ cells from our 18 normal-karyotype MDS cases, using a k-nearest neighbor (KNN) classifier and Euclidean distance (k = 3) to predict the class label by a majority vote. 10 of these cases (56%) were classified as *MYBL2* low, while the remaining eight cases had an *MYBL2* signature that was statistically similar to that of normal CD34+ controls (*Figure 3—figure supplement 1*). To validate our classification algorithm using a separate dataset, we then interrogated the normal karyotype cases (n = 94) reported by Pellagati et al. (*Pellagatti et al., 2010*), identifying a *MYBL2*-low signature in 36 cases (38%), most of which (30 cases) were identified at a confidence level of 1.0. 58 cases (62%) had a normal MYBL2 signature (*Figure 3*).

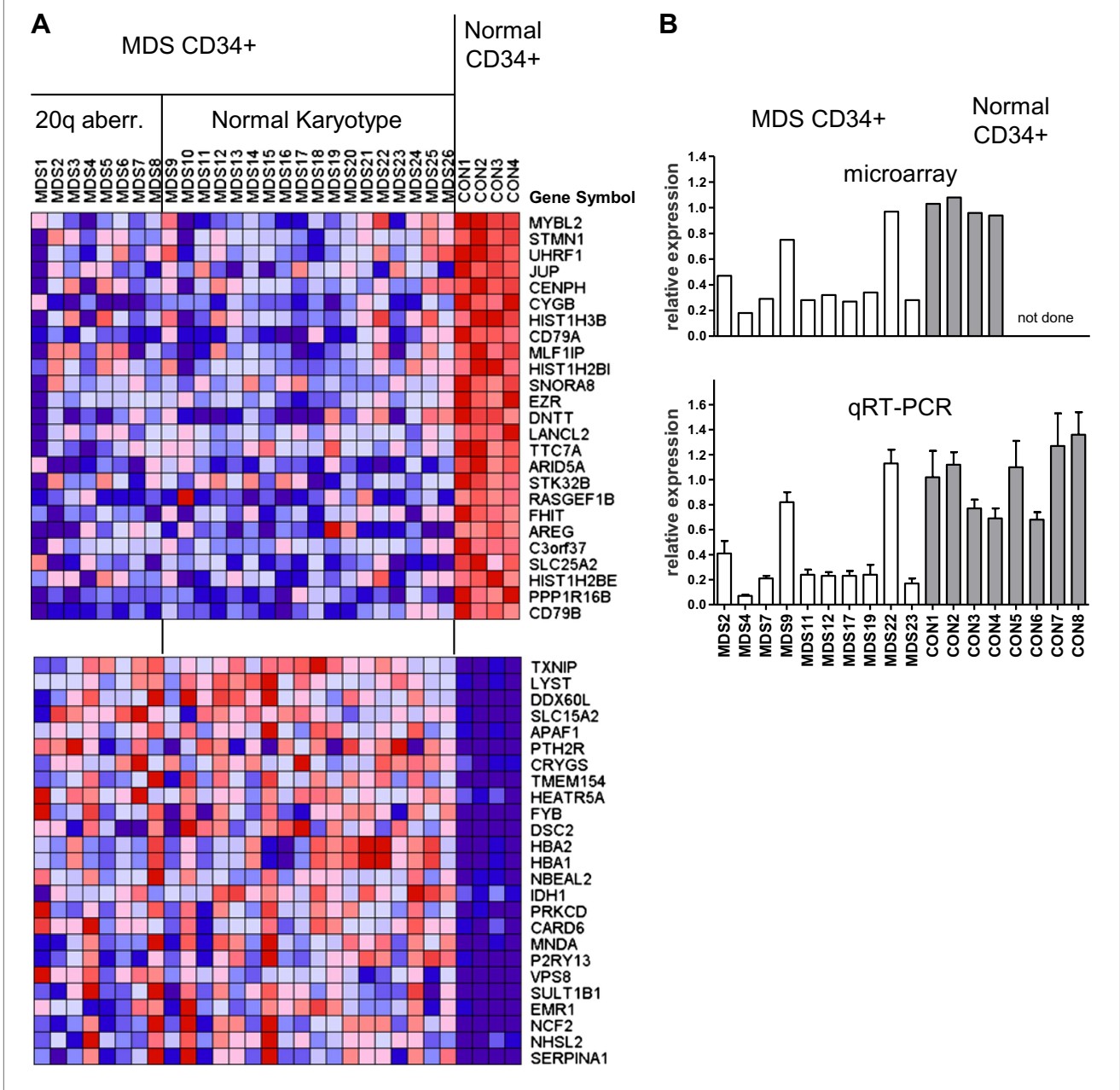

**Figure 1**. Gene expression analysis of CD34+ cells from MDS patients with a 20q aberration or a normal karyotype. (**A**) Heat map showing the top-scoring 25 genes that were either up- or downregulated in CD34+ cells from 26 MDS patients vs normal donors (see ***Supplementary file 1A*** for patient characteristics). Genes were considered differentially expressed if they met two criteria: q ≤ 0.006 and fold-change ≥2.0 (108 genes). (**B**) Validation of microarray-based *MYBL2* gene expression data for selected clinical samples (upper panel) by qRT-PCR analysis (lower panel), including four additional samples of normal CD34+ bone marrow cells. Relative *MYBL2* expression results by qRT-PCR analysis were normalized to three control genes.

The following figure supplements are available for figure 1:

**Figure supplement 1**. Gene expression analysis comparing CD34+ MDS cells with a 20q aberration to normal CD34+ cells.

**Figure supplement 2**. Genomic map of the chromosome 20 CDR.

**Figure supplement 3**. *MYBL2* expression levels.

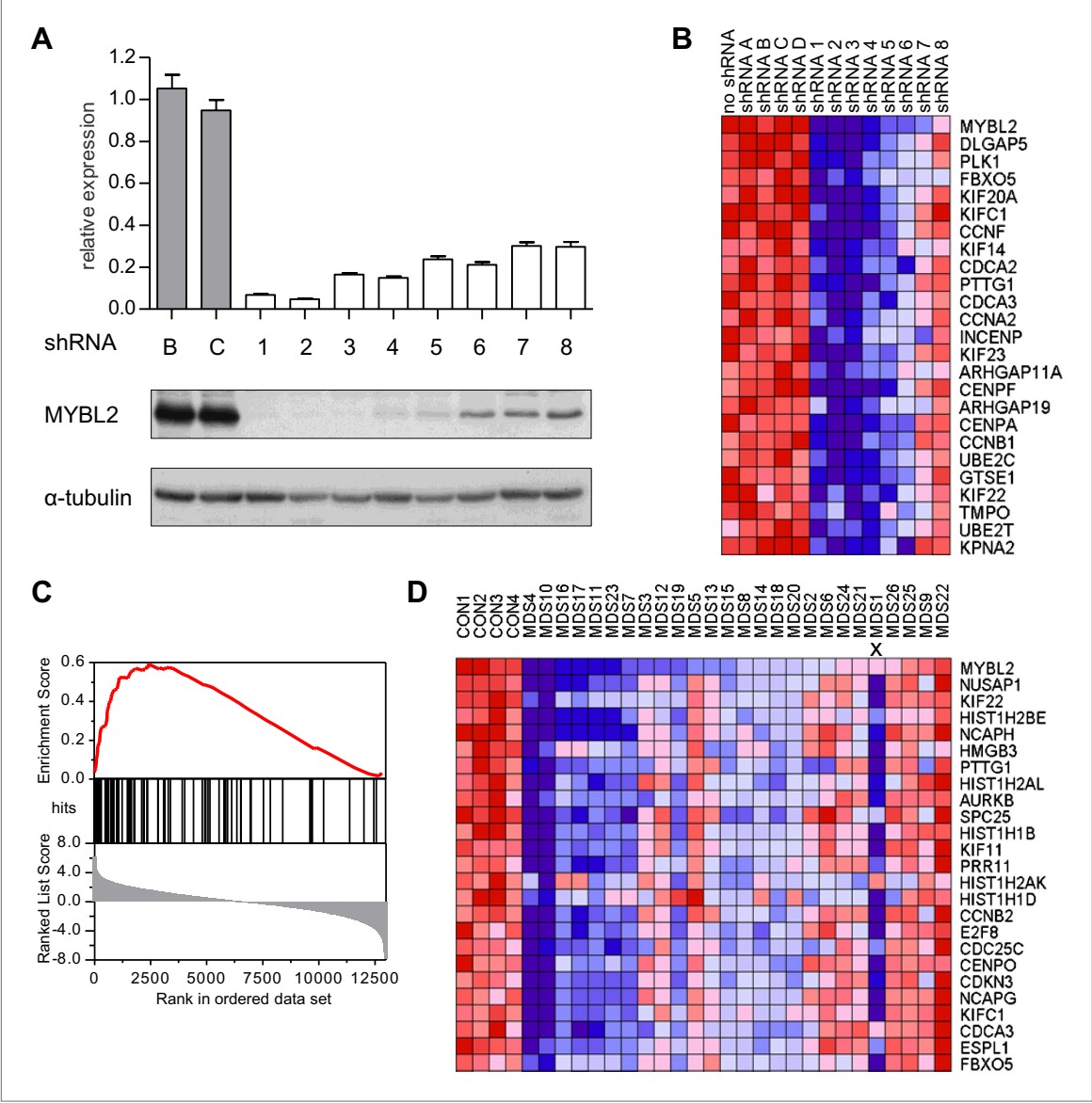

**Figure 2**. Identification of an *MYBL2*-regulated gene expression signature and its enrichment in CD34+ cells from MDS patients. (**A**) *MYBL2* knockdown in SKM1 cells by a series of shRNAs (control shRNAs are labeled B and C; *MYBL2*-specific shRNAs are labeled 1–8). *MYBL2* expression levels were measured by qRT-PCR normalized to three control genes (upper panel) and by Western blotting with tubulin as a normalization control (lower panels). (**B**) Heat map of the top 25 genes whose expression levels positively correlated (score ≥ 0.8) with those of *MYBL2*. Gene expression in normal CD34+ cells was measured by microarray in one unperturbed sample ('no shRNA'), four control shRNA-expressing samples (shRNAs **A–D**) and eight *MYBL2*-specific shRNA-expressing samples (shRNAs 1-8). (**C**) Gene set enrichment analysis (GSEA) of the *MYBL2* gene signature (top 81 positively correlated genes) within the ranked gene expression results for CD34+ cells from MDS patients, compared to CD34+ cells from normal bone marrow. Enrichment of the *MYBL2* signature in MDS bone marrow cells (see **Figure 1A**) is apparent. Genes are ranked by score and plotted on the x-axis; hits are indicated by vertical black bars and the enrichment score is plotted in red. (**D**) Heat map corresponding to the GSEA showing the 25 top-scoring genes. One sample, indicated by an 'x', shows enrichment of the *MYBL2* signature, but relatively normal *MYBL2* expression suggesting a measurement problem.

The following figure supplements are available for figure 2:

**Figure supplement 1**. Gene expression analysis upon *MYBL2* knockdown.

**Figure supplement 2**. Gene set enrichment analysis (GSEA) of the expression profile of CD34+ cells upon *MYBL2* knockdown.

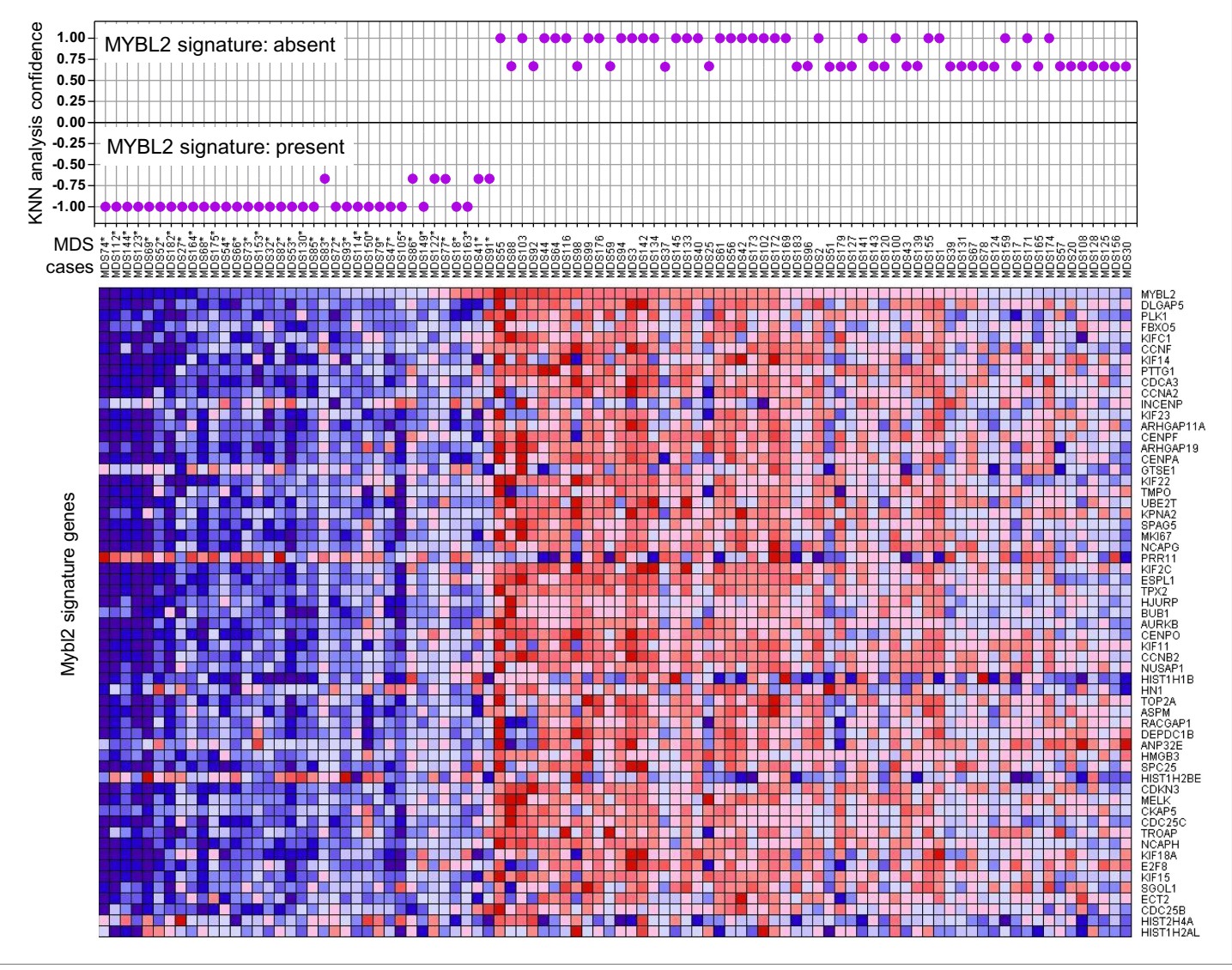

**Figure 3**. KNN classification for normal karyotype patient samples of an independent data set (***Pellagatti et al., 2010***). A k-nearest-neighbor (KNN) classification model for patient samples was trained on our gene expression data set of CD34+ cells with or without MYBL2 knockdown by RNAi. Upper panel: KNN classification for the presence (negative values) or absence (positive values) for the *MYBL2* signature using Euclidean distance (k = 3) to predict the class label by a majority vote. Lower panel: gene expression profile of *MYBL2* signature genes. Note that some gene profiles (INCEP, GTSE1, PRR11, HIST1H1B, HIST1H2AL, HIST1H2BE, HIST2H4A, HN1) appear not to match the overall signatures most likely due to platform inconsistencies.

The following figure supplements are available for figure 3:

**Figure supplement 1**. KNN classification for normal karyotype patient samples.

Thus, we were able to confirm in an independent cohort that the 'MYBL2 low' signature is present in CD34+ cells from a substantial fraction of normal-karyotype MDS cases.

## Mechanisms of *MYBL2* dosage reduction

To begin to clarify the mechanism of reduced MYBL2 activity or aberrantly low *MYBL2* expression in MDS, we resequenced DNA samples from 144 patients, identifying two cases with heterozygous somatic mutations within the coding sequence that were verified by analysis of patient-specific germline DNA collected from buccal swabs (***Figure 4A***). Both mutations, an amino acid substitution and a nonsense mutation, were located within the C-terminus. To test whether these mutations inactivate the MYBL2 protein, we replaced endogenous *MYBL2* in SKM1 cells with *MYBL2* carrying one of the

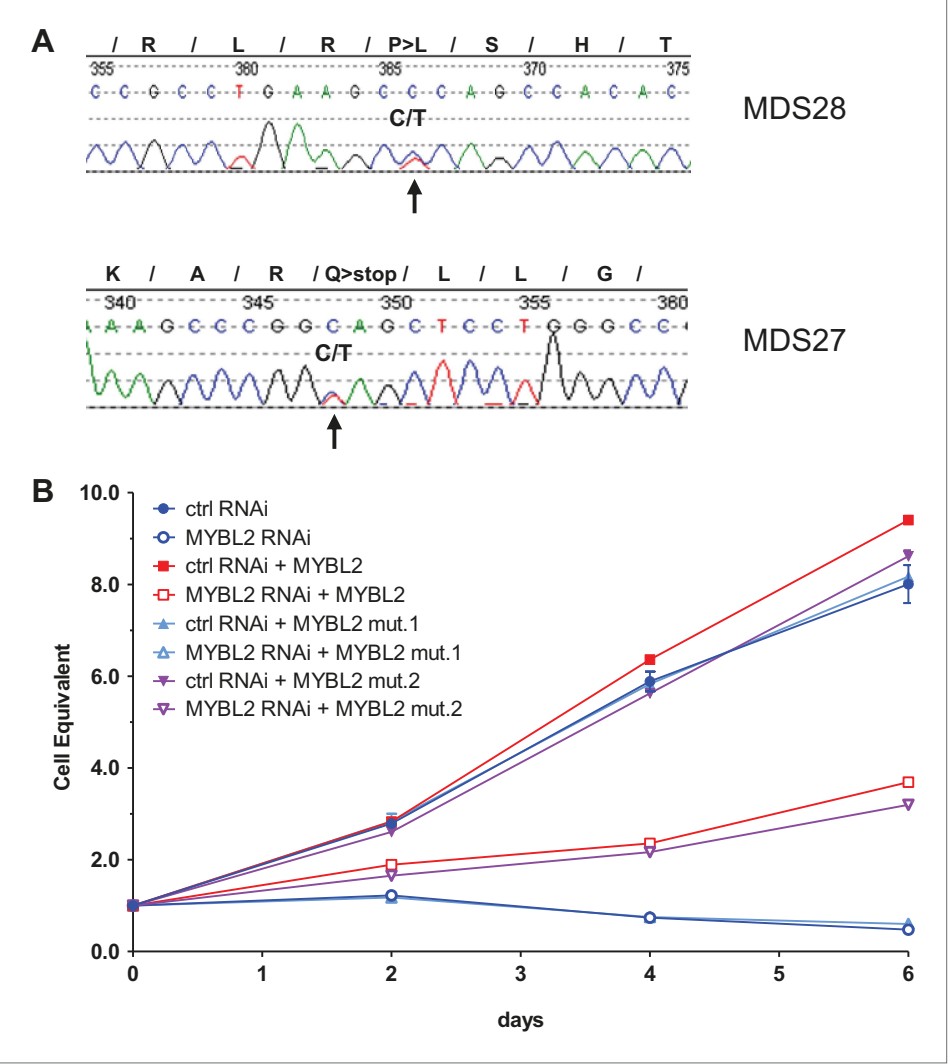

**Figure 4**. *MYBL2* is mutated at a low frequency in MDS. (**A**) *MYBL2* exons were resequenced in the DNA of mononuclear bone marrow cells of 144 MDS patients. The results indicated a P-to-L aminoacid substitution (top, mutation 1, patient MDS28) and a nonsense mutation (bottom, mutation 2, patient MDS27). (**B**) Growth curves of SKM1 cells after transduction with a MYBL2 shRNA. SKM1 cells were transduced with an empty vector (circles), a vector expressing wildtype *MYBL2* cDNA (squares), mutant *MYBL2* cDNA (mut.1, triangles) or mutant *MYBL2* cDNA (mut.2, diamonds). Upon transduction with a control (solid symbols) or a *MYBL2*-specific (open symbols) shRNA-expressing vector growth was monitored for 6 days.

The following figure supplements are available for figure 4:

**Figure supplement 1**. Analysis of *Mybl2* expression in context with mir-29a and mir-30e expression in selected patient samples and controls.

mutations, using a knockdown/rescue approach. SKM1 cells died upon complete knockdown of *MYBL2* with an shRNA targeting the 3′UTR (*Figure 4B*). This phenotype was rescued by re-expression of the wild-type *MYBL2* cDNA and the cDNA from patient 28 (*Figure 4A*), but not by re-expression of the mutated, truncated *MYBL2* cDNA of patient 27, which failed to restore the viability of the SKM1 cells. Thus, we have documented a somatically acquired gene-specific inactivating mutation that targets *MYBL2* in one case of MDS. This specific case is highly informative because the inactivating mutation targets *MYBL2* directly, and not an adjacent gene on 20q.

Given that a substantial number of MYBL2-low cases lacked either a 20q deletion or gene-specific inactivating mutation, we hypothesized that additional mechanisms of *MYBL2* downregulation exist in

MDS. A potential candidate is aberrant DNA methylation, especially in view of the 921 base-pair CpG island that spans the proximal promoter and the first exon of the *MYBL2* gene. However, analysis of this site using bone marrow cell DNA from 30 MDS patients failed to detect any methylated cytosines, indicating that DNA methylation is not a frequent mechanism for *MYBL2* downregulation in MDS (JP Issa, personal communication, 2012). This finding is supported by a recently published study that did not identify methylation affecting the *MYBL2* gene in cells from 83 MDS cases (*Del Rey et al., 2012*).

We also considered that the upregulation of one or more miRNAs targeting *MYBL2* might explain our results. We pursued this notion in a subset of patient samples (seven with low *MYBL2* expression and two with normal *MYBL2* expression, together with three controls) by analyzing the expression levels of *mir-29a* and *mir-30e*. Both of these miRNAs target *MYBL2* (*Han et al., 2010*; *Martinez et al., 2011*), with aberrant expression of *mir-29a* leading to clonal dominance and acute myeloid leukemia (*Han et al., 2010*). Our results show upregulation of either *mir-29a* or *mir-30e* in three cases with *MYBL2* downregulation, but not in controls or cases with normal *MYBL2* expression (*Figure 4—figure supplement 1*). Thus, overexpression of miRNAs targeting MYBL2 affords an attractive mechanism by which MYBL2 expression levels are lowered in a substantial fraction of MDS cases.

## Clonal dominance of transplanted cells with low *Mybl2* expression

If sub-haploinsufficient decreases in *MYBL2* dosage contribute to MDS pathogenesis, a technique such as RNAi knockdown with a series of shRNA hairpins of graded potencies should provide an experimental model that can be used to demonstrate a clonal advantage within hematopoietic progenitor cells and implicate the level of *Mybl2* knockdown that produces the maximum effect. Thus, using five independent, specific shRNA-expressing vectors to target *Mybl2* in primitive hematopoietic cells, we established a competitive reconstitution assay in mice. These vectors downregulated *Mybl2* expression to levels that ranged from 13% (M1) to 33% (M5) of those expressed by control shRNA-expressing vectors in 32D cells, a murine myeloid cell line (*Figure 5—figure supplement 1*). After transducing Lin⁻ mononuclear bone marrow cells in separate vials with these five vectors, which coexpressed a green fluorescent protein (GFP), or with two different control shRNA vectors coexpressing a red fluorescent protein (RFP) (*Strack et al., 2008*), we washed the cells, pooled the five *Mybl2*-specific shRNA knockdown cell aliquots and the two control aliquots, and transplanted equal numbers of these two cell pools into lethally irradiated mice (*Figure 5A*). A second group of mice were transplanted with pooled GFP+ and RFP+ cells expressing control shRNAs only, generating control/GFP vs control/RFP recipients (*Figure 5—figure supplement 2A*). Analysis of GFP and RFP levels in peripheral blood at 12 weeks (*Figure 5B*) showed a strikingly higher median frequency of GFP+ vs RFP+ cells in the peripheral blood of the mice reconstituted with GFP+/*Mybl2*-shRNA vs RFP+/control shRNA (30.9% vs 2.6%, p<0.0001). By contrast, in mice injected with control/GFP vs control/RFP cells, the median frequencies of both RFP- and GFP-expressing cells were low (1.5% and 3.4%; *Figure 5—figure supplement 2B*).

The possibility that these results were influenced by the use of pooled cells representing different shRNA sequences was excluded by transducing the Lin⁻ cells with single hairpins specific for *Mybl2*. For this experiment (*Figure 6A*) we transduced Lin⁻ bone marrow cells with either the M3 shRNA vector (n = 7 mice) or the M4 shRNA vector (n = 8 mice). These two shRNAs were chosen because QPCR analysis of the mice injected with pools of cells transduced with each of the five shRNAs (*Figure 5A*) showed that M3 and M4 were integrated into the DNA of the reconstituted cells that exhibited clonal dominance (*Figure 6—figure supplement 1*). The results of transplanting cells transduced with single *Mybl2*-specific hairpins clearly show clonal dominance by cells transduced with either of these vectors (*Figure 6B*). In a 'color-swap' experiment using the M3 shRNA, the bone marrow cells with *Mybl2* knockdown became clonally dominant, whether the red or green fluorophore was used (*Figure 6C*). Together, these competitive reconstitution studies firmly establish the surprising finding that downregulation of *Mybl2* levels to ~30% of normal levels using two different *Mybl2* shRNAs imparts a strong clonal advantage to hematopoietic progenitor cells, one that becomes reflected in the analysis of circulating white blood cells as early as 12 weeks after transplantation.

## Progression of *Mybl2-low* cells to myeloid malignancy

The marked competitive expansion of *Mybl2* knockdown cells could represent either a benign process or a myeloid malignancy. Thus, we sacrificed 17 mice (nine transplanted with pooled cells

eLIFE Research article

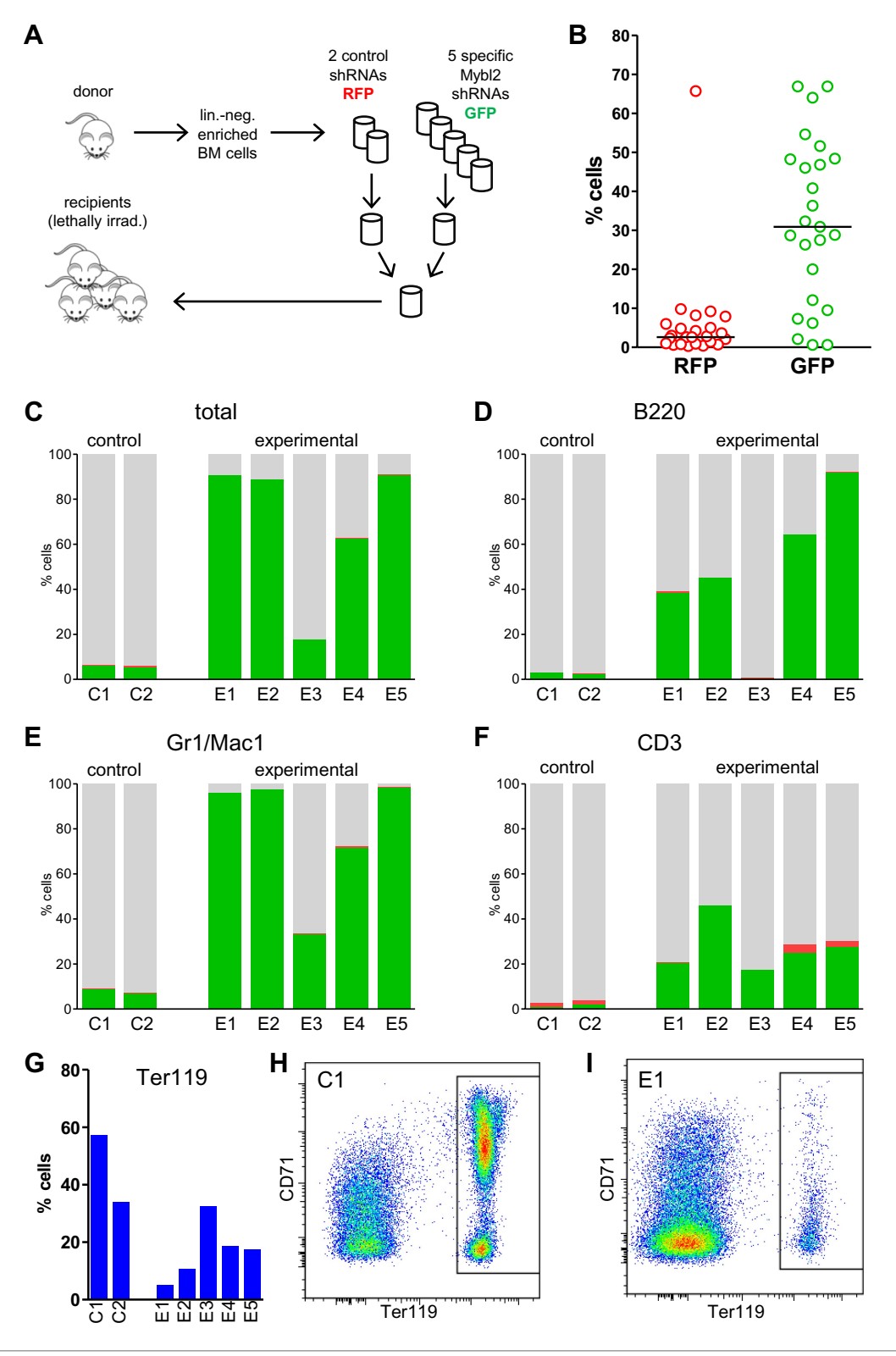

**Figure 5**. Competitive in vivo reconstitution assay testing expansion capacity of cells with low *Mybl2* expression. (**A**) Mononuclear bone marrow cells were collected from donor mice, enriched for lineage-negative cells and transduced in separate wells with specific *Mybl2* shRNA/*GFP* vectors and control shRNA/*RFP* vectors. Equal numbers of cells

*Figure 5. Continued on next page*

*Figure 5. Continued*

were combined to obtain a pool with specific shRNA-expressing cells and a pool of control shRNA-expressing cells. Both pools were combined in equal parts and transplanted into lethally irradiated mice (n = 25) in two independent experiments with transduction efficiencies of 5–8%. (**B**) Analysis of peripheral blood mononuclear cells for GFP and RFP positivity at 12 weeks post-transplant as an indicator of long-term reconstitution (***Morrison and Weissman, 1994***) (GFP+ vs RFP+, p<0.0001 by paired *t*-test; horizontal bars denote median values). (**C–F**), Analysis of bone marrow samples from mice 6–7 months after competitive reconstitution. Flow cytometric analysis of mononuclear bone marrow cells from five experimental mice (E1–E5) transplanted with *Mybl2*-specific shRNA/*GFP* cells and control shRNA/*RFP* cells showed marked overrepresentation of GFP+ cells by comparison with results for control mice (C1, C2) transplanted with control shRNA/*RFP*- and shRNA/*GFP*-transduced cells. The fractions of GFP (green), RFP (red) and unlabeled cells (gray) are shown for the total cell populations (**C**), B220-positive population (**D**), Gr1/Mac1-positive population (**E**), and CD3-positive population (**F**). (**G**) Respective analysis of bone marrow erythropoiesis by CD71/Ter119 staining. Frequencies of Ter119-positive cells are shown for all animals, and representative plots presented for mouse C1 (**H**) and mouse E1 (**I**).

The following figure supplements are available for figure 5:

**Figure supplement 1**. *Mybl2* knockdown in 32D cells by a series of shRNAs.

**Figure supplement 2**. Control experiment for the competitive in vivo reconstitution assay testing expansion capacity of cells with low *Mybl2* expression.

**Figure supplement 3**. Analysis of bood samples from mice 6–7 months after competitive reconstitution.

---

[E1–E5, PD1–PD5] plus three controls and eight transplanted with cells transduced with a single vector [E6-E13] plus two controls) at 6 to 7 months post-transplantation, and studied their bone marrow cells by flow cytometry.

The results confirmed expansion of the *Mybl2*-knockdown cell population compared to control bone marrow cells (***Figures 5C and 6D***; ***Figure 8—figure supplement 1C***), which surpassed even the percentage of GFP+ cells in the blood (***Figure 5—figure supplement 3***). Lineage analysis showed that this expansion included lymphocytes as well as myeloid cells (***Figure 5D–F***), indicating clonal dominance originating from transduced multilineage long-term repopulating cells. Cells of the myeloid lineage were most prominently affected, as the frequency of GFP+ cells in the Gr1+/Mac1+ population exceeded 90% in three of five experimental animals. Bone marrow erythropoiesis was reduced by *Mybl2* knockdown, as indicated by the decreased percentages of Ter119+ cells (***Figures 5G–I and 6E***; ***Figure 8—figure supplement 1D***). As expected, analysis by Western blotting and QRT-PCR of bone marrow cells expressing the M3 *Mybl2*-shRNA as a single hairpin showed that *Mybl2* expression levels were reduced to approximately 20–30% of control levels after long-term reconstitution (***Figure 6F***). We did not observe any effect of this level of *Mybl2* knockdown on the cell cycle-phase distribution of bone marrow cells from mice 6–7 months post-transplantation (***Figure 6—figure supplement 2***), indicating that this level of decreased Mybl2 expression does not induce a delay in the G2-to-M-phase transition.

We also evaluated the circulating blood counts in 13 of the 17 mice at the time of euthanasia 6–7 months after transplant. Reduced hemoglobin levels indicating anemia were observed in 12 of the 13 mice (***Figure 7A***), indicative of the ineffective erythropoiesis in the bone marrow observed by flow cytometry. Circulating white blood cell and platelet counts were in the normal range. Spleen weight was increased in 12 of the 13 mice, reflecting extramedullary hematopoiesis. A few of the mice showed hunched posture, weight loss and lethargy at the time of euthanasia (6–7 months post-transplant), but the majority of the recipient mice appeared well with no evidence of overt hematologic disease.

Histopathologic analysis of the spleen (***Figure 7B–E***) and the bone marrow (***Figure 7F–K***) at 6 to 7 months post transplantation showed marked changes in the overall tissue architecture and in specific cellular constituents. The bone marrow in experimental mice was hypercellular and contained more than 95% mature myeloid cells at the expense of erythroid cells (<5%), while 2% of the cells had a dysplastic morphology. The spleens were frequently enlarged (***Figure 7A***), the white pulp was virtually absent, and the whole organ showed evidence of extramedullary hematopoiesis. Notably, one mouse developed an erythroid leukemia characterized by a block of differentiation at the CD71-positive

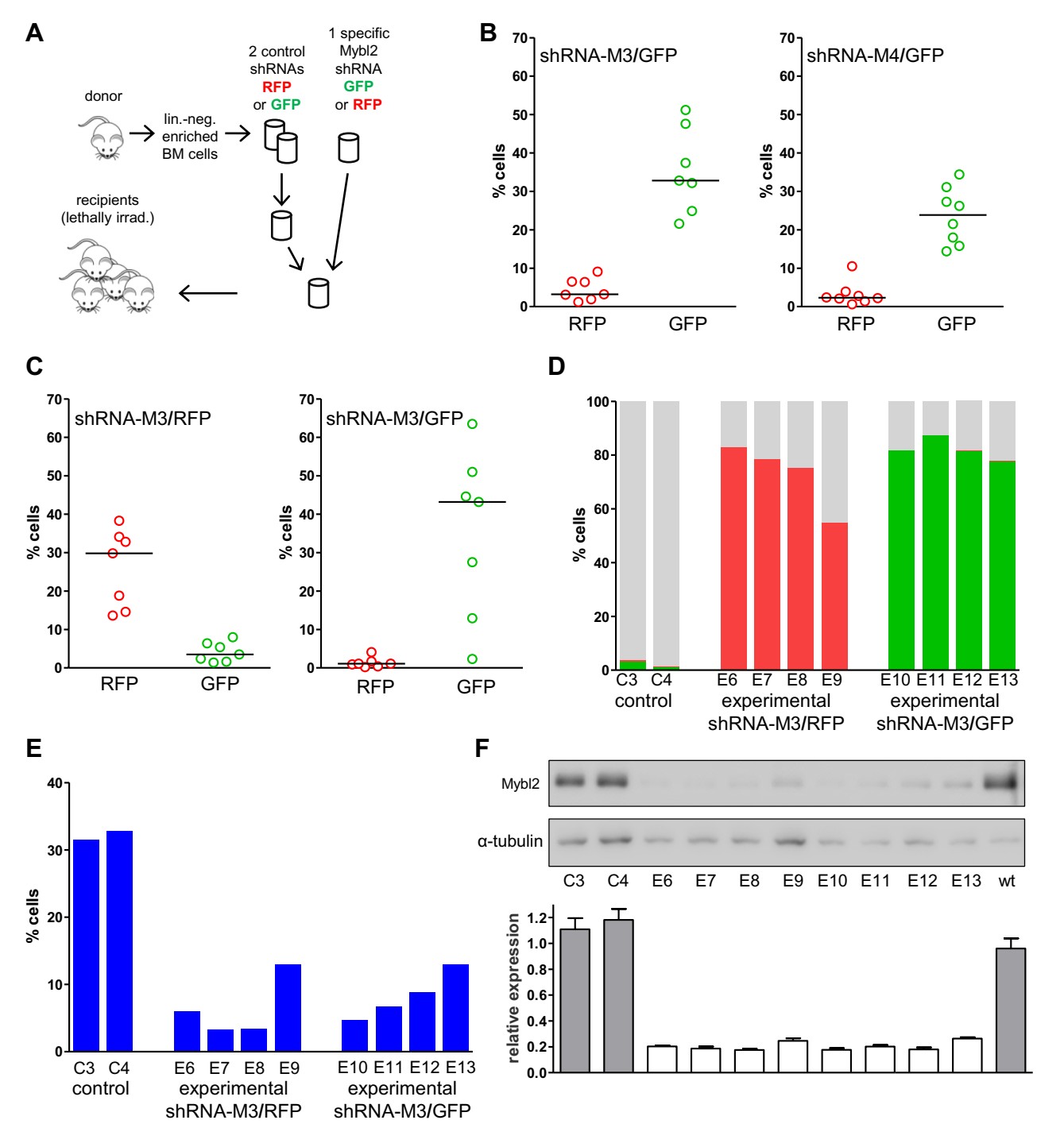

**Figure 6**. Analysis of peripheral blood and bone marrow samples from mice transplanted with single *Mybl2*-targeting shRNA vectors. (**A**) Mononuclear bone marrow cells were collected from donor mice, enriched for lineage-negative cells and transduced in separate wells with a single, specific *Mybl2* shRNA vector or control shRNA vector with an efficiency of <10%. The vectors allowed for co-expression of *RFP* or *GFP*. Equal numbers of control shRNA-expressing cells and *Mybl2*-specific shRNA-expressing cells were combined and transplanted into lethally irradiated mice in two independent experiments. (**B**) Experiment 1: Analysis of peripheral blood mononuclear cells for GFP and RFP positivity at 12 weeks post-transplant as an indicator of long-term reconstitution. Significant expansion of GFP-positive cells with use of either shRNA M3 (left, p=0.0004 by paired *t*-test) or M4 (right, p=0.0005). Horizontal bars denote median values. (**C**) Experiment 2, 'color-swap' experiment: *Mybl2*-specific shRNA M3 coupled with *GFP* (right) or *RFP* (left) expression associated with a significant expansion of *Mybl2*-downregulated cells (p=0.007 and p=0.001, respectively). (**D**) Flow cytometric analysis of
*Figure 6. Continued on next page*

*Figure 6. Continued*

mononuclear bone marrow cells from eight experimental mice (E6–E13) transplanted with cells expressing *Mybl2*-specific M3 shRNA/*GFP* and control shRNA/*RFP* cells (E6–E9) or *Mybl2*-specific M3 shRNA/*RFP* cells and control shRNA/*GFP* cells (E10–E13). Mybl2-RNAi cells were markedly overrepresented by comparison to results for control mice (C3, C4) transplanted with control shRNA/*RFP*- and shRNA/*GFP*-transduced cells. The fractions of *GFP* (green), *RPF* (red) and unlabeled cells (gray) are shown. (**E**) Analysis of bone marrow erythropoiesis by CD71/Ter119 staining with frequencies of Ter119-positive cell frequencies shown for all animals. (**F**) Analysis of Mybl2 levels of flow sorted cells by Western-blotting (upper panel) and qRT-PCR (lower panel).

The following figure supplements are available for figure 6:

**Figure supplement 1**. Analysis of genomic DNA for the presence of specific *Mybl2*-targeting shRNA sequences.

**Figure supplement 2**. Cell cycle analysis.

stage at 6 months post-transplantation (***Figure 7—figure supplement 1***). In summary, the histopathologic findings in these mice indicate a clonal myeloproliferative disorder with dysplastic features.

To identify the cells principally involved in the clonal dominance associated with this disorder, we analyzed the Lin⁻/Sca1⁺/Kit⁺ (LSK) bone marrow cells of the affected mice (n = 12) in comparison to controls. Within the LSK population of GFP+ or RFP+ *Mybl2*-knockdown bone marrow cells, HSCs (CD150+/CD48−) were diminished, MPPs (CD150−/CD48−) were maintained and LRPs (CD150−/CD48+) were increased compared to unlabeled, non-transduced cells. Thus, the clonal dominance due to low *Mybl2* levels is first evident as an increased percentage of cells in the LRP fraction. (***Figure 7—figure supplement 2***).

We next performed secondary transplantation experiments to determine if self-renewing neoplastic cells were involved. We first analyzed four primary donors in detail and confirmed the presence of clonally dominant GFP+/*Mybl2*-RNAi cells in the peripheral blood and bone marrow of these mice (***Figure 8—figure supplement 1A,C***). Moreover, the mice were anemic based on low RBC and low hemoglobin levels (***Figure 8—figure supplement 1B***) and had markedly reduced erythropoiesis, as shown by low percentages of CD71neg/Ter119+ cells compared to that in a control transduced mouse (***Figure 8—figure supplement 1D***). For the secondary transplants, we injected 15 mice with bone marrow cells and eight with spleen cells that originated from these four individual donors (no pooling).

Analysis of blood samples collected from secondary recipients at 10 weeks revealed a marked clonal advantage for GFP+/Mybl2-RNAi cells transplanted from each of the four mice with primary disease, regardless of whether the cell source was spleen or bone marrow (***Figure 8A***). In addition, there was evidence of anemia (low RBC and low Hb levels) and thrombocytopenia in 9 of 13 animals (***Figure 8B***). The bone marrow of secondary recipients was also dominated by GFP+/Mybl2-RNAi cells (***Figure 8C***) and showed decreased levels of erythropoiesis (***Figure 8D***) as well as histologic evidence of a myeloproliferative disorder with dysplasia. Overall, the phenotypic consequences of Mybl2 knockdown were much more pronounced in secondary compared to primary recipients.

Thus, our secondary transplantation assays show that either spleen or bone marrow cells from mice with myeloid neoplasia due to shRNA-mediated reduced *Mybl2* expression can efficiently transmit the blood disorder to secondary recipients. This indicates that the disorder is cell autonomous and that hematopoietic progenitor cells with reduced *Mybl2* expression can self-renew after transplantation and undergo sustained expansion leading to clonal dominance.

## Discussion

Recent molecular studies of MDS have uncovered a number of previously unrecognized genes whose aberrant inactivation through mutation or other means can disrupt hematopoietic cell development, leading to ineffective hematopoiesis and a myelodysplastic phenotype (***Shih et al., 2012***). The picture beginning to emerge from these investigations is that a broad array of molecular changes have the potential to drive the biology and clinical phenotypes of MDS in unique ways. Yet, the disease remains very heterogeneous and none of the molecular lesions that have been convincingly linked to MDS pathogenesis affect more than a fourth of cases overall, with a prevalence of 1–15% being much more common (***Bejar et al., 2011***; ***Shih et al., 2012***).

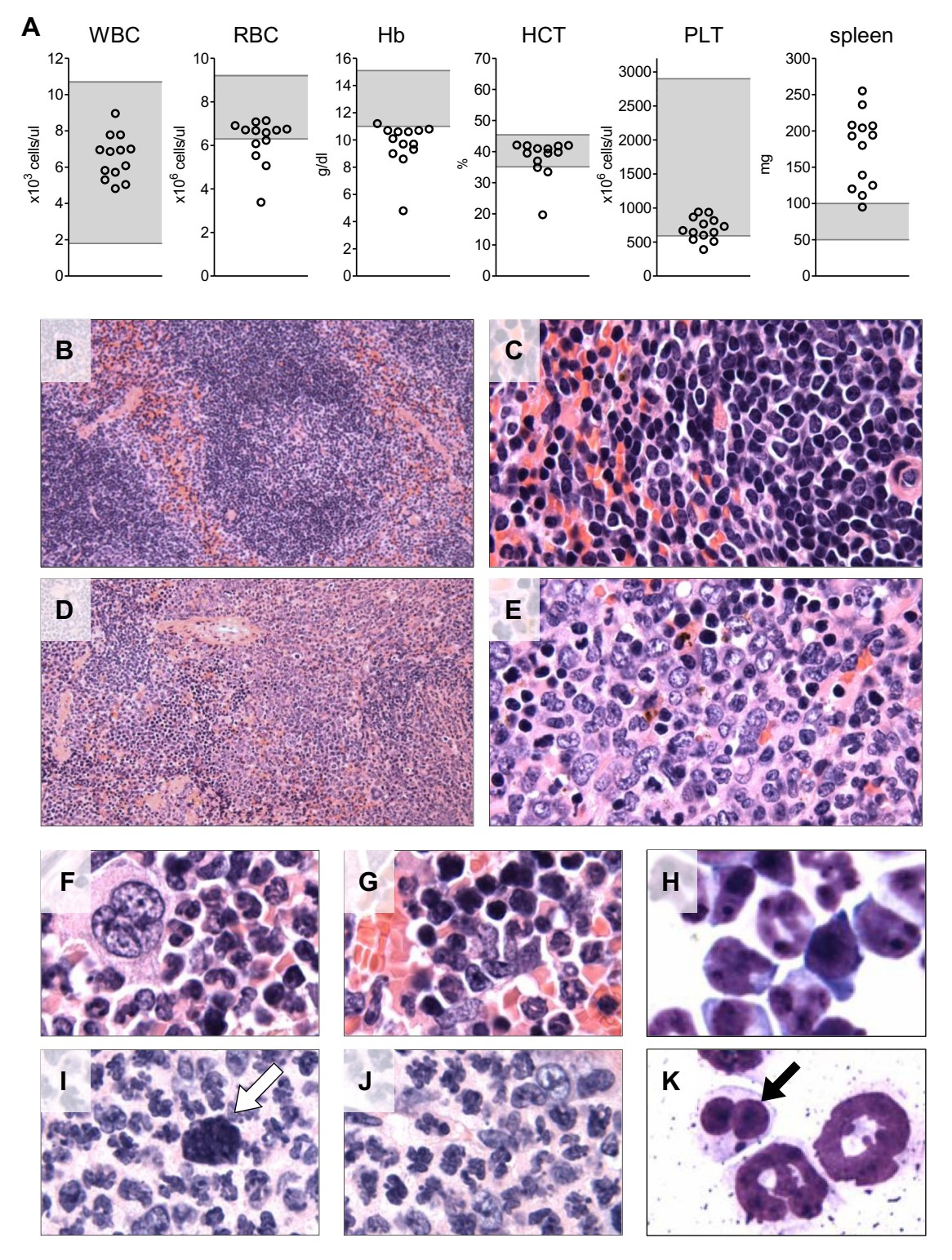

**Figure 7**. Analysis of blood counts and spleen weight (A), spleen (B–E) and bone marrow (F–K) samples from mice 6-7 months after competitive reconstitution. (**A**) Analysis of peripheral blood parameters showing anemia of animals that received transplants of specific *Mybl2*-shRNA engineered cells: white blood cell count (WBC), red blood cell count (RBC), hemoglobin (HB), hematocrit (HCT), and platelet count (PLT). Spleen weights are shown as well. Grey areas indicate the normal range for each value. (**B** and **C**) Normal spleen histology at different magnification of control mouse C1 showing normal lymphoid-rich white pulp (50% spleen volume) and red pulp (50%). About 10% of the spleen consists of extramedullary hematopoiesis, which is in

*Figure 7. Continued on next page*

*Figure 7. Continued*

the red pulp and normal for mice. (**D** and **E**) Spleen histology of mouse E1 with specific *Mybl2*-knockdown at different magnification shows a marked depletion of white pulp. About 90% consists of red pulp, which is almost entirely extramedullary hematopoiesis. (**F**–**K**) Normal bone marrow histology (**F** and **G**) and cytology (**H**) of control mouse C1, compared to corresponding results for mouse E1 with specific *Mybl2*-knockdown (histology **I**, **J**; cytology **K**). Note marked shift to myeloid elements with loss of erythropoiesis and occasional dysplastic myeloid cells (white arrow: hypolobate megakaryocyte, black arrow: dysplastic neutrophil).

The following figure supplements are available for figure 7:

**Figure supplement 1**. Erythroid leukemia associated with *Mybl2* knockdown.

**Figure supplement 2**. Flow cytometric analysis of the LSK population in the bone marrow of mice upon reconstitution with *Mybl2* shRNA-expressing cells.

In this study, we identified *MYBL2* as a dosage-dependent tumor suppressor gene that was reduced in expression to sub-haploinsufficient levels in 17 (65%) of 26 human MDS cases. This result is somewhat surprising given that *MYBL2* has been implicated as a gene that is typically overexpressed (rather than downregulated) in several types of cancer and whose elevated expression correlates with a poor prognosis in human breast cancer and other malignancies (*Amatschek et al., 2004*; *Paik et al., 2004*). However, no direct evidence demonstrating oncogenic activity by over-expressed *MYBL2* has been reported to date. High levels of expression of this transcription factor in other tumor types may reflect the greater proliferative rate of the tumor cell population compared to resting cells, so that the apparently levels of high *MYBL2* activity may be a consequence rather than a cause of malignant transformation. By contrast, the proliferative fraction of cells in myeloid neoplasia is generally quite low.

Our findings demonstrate the surprising finding that abnormally low levels of *MYBL2* expression have the unique property of imparting a clonal advantage to multipotent hematopoietic progenitors, leading rapidly to the clonal dominance of such cells within the myeloid, and B- and T-lymphocyte cell lineages. The persistence of clonal dominance in recipient mice for more than 6 months after bone marrow cell transplantation, and in recipients of secondary transplants, indicates that the affected multipotent hematopoietic cells have both self-renewal and long-term repopulating ability (*Morrison and Weissman, 1994*). Hence, our finding establishes MYBL2 as a key tumor suppressor gene of the 20q CDR affected in human MDS and MPD. However, we cannot exclude the possibility that haploinsufficiency for other genes within the 20q CDR cooperates with aberrantly low expression of *MYBL2* to move the cells toward malignant transformation, as has been shown for the CDR associated with the 5q-syndrome, where haploinsufficiency of the gene *RPS14* as well as loss of expression of an miRNA (*Ebert et al., 2008*; *Barlow et al., 2010*; *Starczynowski et al., 2010*) are critical steps in MDS pathogenesis.

Most intriguing is our finding of decreased *MYBL2* expression in MDS cases with a normal karyotype, which suggests that interstitial deletion of chromosome 20q is not the only mechanism by which *MYBL2* activity can be reduced in MDS. Indeed, low levels of *MYBL2* were apparent in 10 (56%) of 18 of our normal-karyotype MDS cases, and 38% of those reported by Pellagatti et al. (*Pellagatti et al., 2010*). We were unable to implicate aberrant methylation of the *MYBL2* genomic locus as an additional mechanism of gene-dosage reduction; instead, we found elevated expression levels of *mir-29a* and *mir-30e*, which are known to target *MYBL2* (*Martinez et al., 2011*), in CD34+ cells from patients with MDS and low levels of *MYBL2* expression. In addition, others have shown that overexpression of *mir-29a* in murine bone marrow cells leads to acute myeloid leukemia (*Han et al., 2010*). Combined with our results, this observation suggests that mir-29a may be oncogenic, at least in part, because it targets *Mybl2* transcripts. Finally, we identified a case with heterozygous missense mutation in *MYBL2* that led to the loss of gene function, analogous to the effect of monoallelic gene deletion. Thus, our results indicate that the MYBL2 transcription factor gene is targeted by diverse molecular events in MDS, including not only deletion within the 20q chromosomal region, but also heterozygous loss-of-function mutations and the repression of gene expression by dysregulated miRNAs.

Clarke et al. (*Clarke et al., 2012*) have also described an association between *MYBL2* expression levels and the occurrence of hematologic disorders, including MDS, in aged *Mybl2* haploinsufficient mice. These authors were able to show that haploinsufficiency for the *Mybl2* gene increases susceptibility

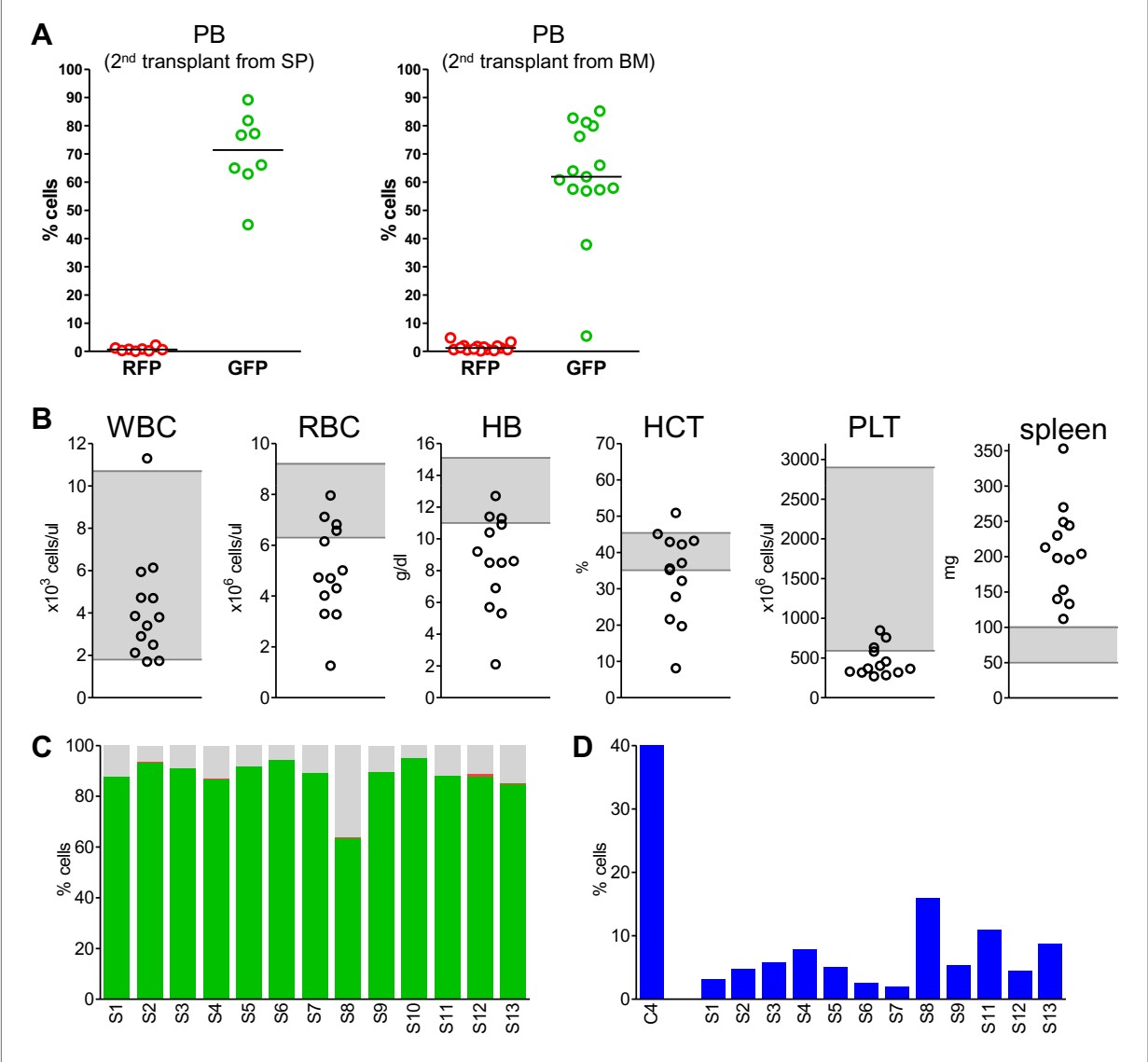

**Figure 8**. Secondary transplants: In vivo reconstitution assay testing capacity of cells with low *Mybl2* expression to engraft in lethally irradiated mice. (**A**) Analysis of peripheral blood mononuclear cells for GFP and RFP positivity at 10 weeks post-transplant showing engraftment of cells originating from spleen (p<0.0001, left panel) or bone marrow cells (p<0.0001, right panel). Horizontal bars denote median values. (**B**) Analysis of peripheral blood parameters showing anemia of animals that received secondary transplants of specific *Mybl2*-shRNA engineered cells: white blood cell count (WBC), red blood cell count (RBC), hemoglobin (HB), hematocrit (HCT), and platelet count (PLT). Spleen weights are shown as well. Grey areas indicate the normal range for each value. (**C**) Flow cytometric analysis of mononuclear bone marrow cells 3 months after transplantation from 13 mice (S1–S13) demonstrating strong engraftment and expansion of cells with a specific *Mybl2* knockdown (color code as in *Figure 5*). (**D**) Analysis of bone marrow erythropoiesis by CD71/Ter119 staining showing the frequencies of CD71−/Ter119+ cells and revealing a strong reduction of Ter119+ cells in all experimental animals in comparison to a wild-type control animal.

The following figure supplements are available for figure 8:

**Figure supplement 1**. Analysis of the peripheral blood and bone marrow 8 months after transplantation of lineage-negative bone marrow cells transduced with *Mybl2*-targeting shRNA expression vectors (primary transplant donors).

to age-related myeloid neoplasia. Our data indicate that more pronounced gene dosage reduction of *Mybl2* might be even more advantageous for clonal dominance. Although our findings do not rule out a causal role for straight-forward haploinsufficiency in myeloid malignancies, they strongly suggest that the oncogenic effects of reduced *MYBL2* dosage are more pronounced as the dosage level

falls to near 30% of normal, in keeping with the continuum model of tumor suppression (***Berger et al., 2011***). We would also stress that in the range of *MYBL2* dosage reductions in this study were modeled based on those found in CD34+ cells from MDS clinical samples, in which the reductions of *MYBL2* gene dosage were predominately in the range of 20–30% of normal levels, and thus were considerably lower than the 50% predicted by classical haploinsufficiency. Importantly, expression levels lower than 20% appear incompatible with cell growth (***Figure 4B***), consistent with the known function of MYBL2 as a key regulator of G2-to-M transition and the fact that biallelic inactivation of this gene causes early embryonic lethality in the mouse due to a block in inner cell mass formation during embryonic development (***Tanaka et al., 1999***).

Our results implicating a transforming role for sub-haploinsufficient levels of *MYBL2* build upon earlier studies implicating progressive inactivation of *CTNNA1* and the graded reduction of *PU.1* expression in the molecular pathogenesis of myeloid malignancies (***Rosenbauer et al., 2004***; ***Liu et al., 2007***). Our approach, which tested the effects of graded levels of *Mybl2* knockdown of hematopoietic progenitors in vivo, provides compelling evidence for the importance of reductions of tumor suppressor gene levels below the 50% level associated with classical haploinsufficiency in the molecular pathogenesis of myeloid malignancies.

## Materials and methods

### Clinical samples

Bone marrow samples represented 28 MDS patients (***Supplementary file 1A***) seen at the MD Anderson Cancer Center (MDACC). The diagnosis and classification of MDS followed criteria of the World Health Organization (WHO), the French-American–British Cooperative Group (FAB) and the International Prognostic Scoring System (IPSS). Mononuclear cells were isolated from bone marrow aspirates by Ficoll centrifugation, and further purified by magnetic beads (Miltenyi Biotec, Auburn, CA) to obtain the CD34+ fraction. Normal CD34+ cells were processed in exactly the same manner.

### Competitive reconstitution assays

Mononuclear cells were isolated from the femurs and tibias of syngeneic mice (C57BL/6), and were depleted of lineage-marked cells using sheep anti-rat IgG-coupled Dynabeads (Life Technologies, Carlsbad, CA) together with affinity-purified rat anti-mouse antibodies (eBioscience, San Diego, CA) against CD3, CD5, B220, Gr1, Mac1, Ter119. Cells were cultured for 24 hr in StemSpan SFEM (Stem Cell Technologies, Vancouver, BC, Canada) supplemented with 100 U/ml penicillin/streptomycin, 2 mM glutamine, 80 ng/ml Scf, 40 ng/ml TPO and 40 ng/ml Flt3l. Cells were aliquoted and transduced in separate wells with single shRNA- or control shRNA-expressing vectors, extensively washed and cultured separately for an additional 24 hr before transplantation. Single vector transduced cells were pooled according to the scheme in ***Figure 5*** just before transplantation. Recipient C57BL/6 mice were lethally irradiated (9.5 Gy) at a dose rate of approximately 1 Gy/min, delivered as a single dose. The next day, mice were reconstituted by retro-orbital venous sinus injection of $2 \times 10^5$ lentivirus-transduced cells mixed with a $1 \times 10^5$ freshly isolated whole bone marrow cells for radioprotection.

### RNA preparation and qRT-PCR

RNA was extracted from Trizol (Life Technologies, Carlsbad, CA) according to the manufacturer's instructions, and 500 ng were used for reverse transcription (Quantitect kit, Qiagen, Hilden, Germany). Real-time PCRs were performed on an ABI PRISM 7700 system using SYBR green-based assays with AmpliTaq Gold (Life Technologies). All reactions were performed in triplicate. Quantitative data were calculated from the Ct-values for each reaction using the mean reaction efficiency for each primer pair. Data were normalized to expression levels of *PAPOLA*, *UBQLN1* and *VPS39* (housekeeping genes) and scaled to the mean of the controls. Sequences of primer sets are available in ***Supplementary file 1C***.

### Microarray analysis

Total RNA was isolated from cells lysed in Trizol (Life Technologies). 100–500 ng of each sample was converted into fragmented, biotinylated cDNA hybridized to a microarray chip (Gene 1.0 ST) and fluorescently labeled according to the standard protocol (Affymetrix, Santa Clara, CA) at the

Dana-Farber microarray core facility. Raw data were processed in Expression Console (Affymetrix) using RMA normalization. Expression values for each gene were annotated by mapping all probe sets to the human genome version hg19. Data complexity was reduced to one canonical transcript per gene, resulting in a single identifier per gene (17,982 total). The expression data were processed in Gene Pattern (*Reich et al., 2006*) (Broad Institute, Cambridge, MA). Non-expressed genes were filtered out, and the resulting expression matrix was analyzed with the comparative marker module (two class comparison with 10,000 permutations) or the gene neighbor module in Gene Pattern. GSEA (Broad Institute) was performed as described previously (*Subramanian et al., 2005*). Data are available at Gene Expression Omnibus, accession number GSE43401.

A k-nearest neighbor (KNN) algorithm was used to classify normal-karyotype MDS cases by using Euclidean distance and k = 3 to predict the class label by a majority vote. To generate a training set, the 89 genes most correlated with *MYBL2* (Pearson correlation coefficient >0.80; 81 positively correlated and 8 negatively correlated) based on the short hairpin knock-down experimental data were reduced to a 59 gene signature positively correlated with *MYBL2* by removing 14 genes which hierarchically clustered with the 8 negatively correlated genes and 9 genes having low expression (≥3 out of 4 controls) in the CD34+ control samples in the 30 patient samples arrays. This signature was also examined in an independent cohort of 94 normal karyotype patients using KNN.

## Western blotting

Protein extracts were prepared by SDS lysis and normalized by total protein content (BCA assay). Total protein (5–25 µg) was separated by SDS polyacrylamide gel electrophoresis and blotted on PVDF membranes. Specific proteins were visualized by chemiluminescence using antibodies for MYBL2 (ab12296, Abcam, Cambridge, United Kingdom) or Tubulin (B-5-1-2, Sigma, St. Louis, MO).

## Cell culture

Cryopreserved human bone marrow CD34+ cells were obtained from Lonza and cultured in StemSpan SFEM (Stem Cell Technologies) supplemented with 100 U/ml penicillin/streptomycin, 2 mM glutamine, 50 µg/ml lipoproteins (EMD Millipore, Billerica, MA), 80 ng/ml SCF, 40 ng/ml TPO, 40 ng/ml FLT3L, 10 ng/ml IL3 and 10 ng/ml IL6. SKM1 cells were obtained from the DSMZ (Braunschweig, Germany) and maintained in RPMI medium supplemented with 10% fetal bovine serum.

## Lentiviral vectors and transduction

Oligonucleotides encoding shRNAs were cloned into pLKO1. pLKO1 derivatives were obtained by replacing the puromycin resistance marker with *EGFP* or *DsRedExpress2* (*Strack et al., 2008*). Lentiviral RNAi vectors are available on request. The *MYBL2* cDNA including a 3xFLAG tag was cloned into derivate of the pLenti6.3 (Life Technologies) vector under control of the *SFFV* promoter. Lentiviral particles were produced in 293T cells by cotransfection of pMD2G, pCMV-dR8.9 and the lentiviral vector. Cells were spin-transduced in the presence of polybrene (4 µg/ml) and selected at 24 hr post-transduction with puromycin (1.5 µg/ml) or blasticidin (3 µg/ml). Growth curves were obtained with the CellTiter-Glo luminescent assay (Promega, Madison, WI).

## Flow cytometry

A FACSCanto II machine (BD Biosciences, San Jose, CA) equipped with a 407 nm, 488 nm and 633 nm laser was used. Staining was performed with the following antibodies (clone id in parentheses), all purchased from eBioscience: Mac1 (M1/70), Gr1 (RB6-8C5), B220 (RA3-6B2), CD3 (145-2C11), CD71 (R17217) and Ter119 (Ter119).

## Competitive reconstitution assays

C57BL/6 mice, purchased from Jackson Laboratories, were housed in individually ventilated cages in the Dana-Farber Cancer Institute animal facility in accord with guidelines of the Institutional Animal Care Use Committee. Transplanted mice received Baytril-containing water (Bayer, Robinson Township, PA) to reduce the probability of infection from opportunistic pathogens in first 30 days after transplantation. For peripheral blood analysis, tailvein blood was collected and RBCs were lysed with ammonium chloride buffer prior to staining for flow cytometry. For bone marrow analysis, cells were isolated

from femurs and tibias with RBCs lysis buffer used for all subsequent procedures except analysis of erythropoiesis by Ter119/CD71 flow cytometry.

## Cytomorphologic and histologic evaluation

Cytospin slides were prepared from isolated mononuclear bone marrow cells and stained with Wright-Giemsa stain. Bone and spleen samples were fixed in 10% formalin, washed with PBS and stored to 80% ethanol. Sections were stained with hematoxylin and eosin.

## Statistical analysis

All data were analyzed with GraphPad Prism, version 5.04 for Windows (http://www.graphpad.com). p-values (two-tailed) for competitive reconstitution were calculated by a paired *t*-test.

## Acknowledgements

We thank John R Gilbert for editorial review.

## Additional information

### Funding

| Funder | Grant reference number | Author |
| --- | --- | --- |
| National Institutes of Health | NIH P01 CA-108631 | Stefan Heinrichs, Lillian F Conover, Carlos E Bueso-Ramos, Outi Kilpivaara, Kristen Stevenson, Donna Neuberg, Mignon L Loh, Guillermo Garcia-Manero, Hagop M Kantarjian |

The funders had no role in study design, data collection and interpretation, or the decision to submit the work for publication.

### Author contributions

SH, Conception and design, Acquisition of data, Analysis and interpretation of data, Drafting or revising the article, Contributed unpublished essential data or reagents; LFC, OK, Acquisition of data, Analysis and interpretation of data; CEB-R, Acquisition of data, Contributed unpublished essential data or reagents; KS, DN, Analysis and interpretation of data, Drafting or revising the article; MLL, Contributed patient samples, Contributed unpublished essential data or reagents; W-SW, Conception and design, Analysis and interpretation of data; SJR, Analysis and interpretation of data; GG-M, HMK, Contributed patient samples, Conception and design; ATL, Conception and design, Analysis and interpretation of data, Drafting or revising the article

### Ethics

Human subjects: The specific ethical approval was obtained for work with human samples from the institutional ethics committee at the M.D. Anderson Cancer Center in accordance with the Declaration of Helsinki by the WMA. Guidelines were strictly followed. Informed consent was obtained for all human samples.

Animal experimentation: Animal care was in strict compliance with institutional guidelines established by the Dana-Farber Cancer Institute, the National Academy of Sciences Guide for the Care and Use of laboratory Animals, and the Association for Assessment and Accreditation of Laboratory Animal Care International. All animal procedures were approved by the Institutional Animal Care and Use Committee (IACUC) at Dana-Farber Cancer Institute (protocol #05-048)

## Additional files

### Supplementary files

• Supplementary file 1. (**A**) MDS patient characteristics. (**B**) Genes positively correlated with MYBL2 (MYBL2 gene signature). (**C**) qRT-PCR primer sequences.

## Major dataset

The following dataset was generated:

| Author(s) | Year | Dataset title | Dataset ID and/or URL | Database, license, and accessibility information |
|---|---|---|---|---|
| Heinrichs S | 2013 | *MYBL2* Is a Sub-haploinsufficient Tumor Suppressor Gene in Myeloid Malignancy | GSE43401; http://www.ncbi.nlm.nih.gov/geo/query/acc.cgi?acc=GSE43401 | Publicly available at GEO (http://www.ncbi.nlm.nih.gov/geo/). |

The following previously published dataset was used:

| Author(s) | Year | Dataset title | Dataset ID and/or URL | Database, license, and accessibility information |
|---|---|---|---|---|
| Pellagatti A, Cazzola M, Giagounidis A, Perry J, et al | 2010 | Expression data from bone marrow CD34+ cells of MDS patients and healthy controls | GSE19429; http://www.ncbi.nlm.nih.gov/geo/query/acc.cgi?acc=GSE19429 | Publicly available at GEO (http://www.ncbi.nlm.nih.gov/geo/). |

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
