## [Decision Letter]

Thank you for sending your work entitled “*MYBL2* Is a Sub-haploinsufficient Tumor Suppressor Gene in Myeloid Malignancy” for consideration at *eLife*. Your article has been favorably evaluated by a Senior editor and 3 reviewers, one of whom, Lou Staudt, is a member of our Board of Reviewing Editors.

The Reviewing editor and the other reviewers discussed their comments before we reached this decision, and the Reviewing editor has assembled the following comments to help you prepare a revised submission.

Major comments:

1) The reviewers are unconvinced by the argument that *MYBL2* levels must be reduced to sub-haploinsufficient levels to cause myeloid malignancy. After all, Clarke et al show that deletion of a single copy of *MYBL2* is sufficient to cause myeloproliferation in mice. The authors do not test an shRNA that only reduces *MYBL2* levels by 50%, so they cannot rule out the possibility that *MYBL2* could function as a typical haploinsufficient tumor suppressor. It would seem best to tone down this aspect of the Discussion.

2) The phenotypic description of the mice shown in Figure 7 is incomplete. First, it is unclear if hematologic disease was the cause of death in any of these mice (except for the one that developed AML) or if these findings were noted at the time of elective euthanasia. A Kaplan–Meier curve showing survival (if reduced) or a clear statement such as “although transduced cells expressing *Mybl2* hairpins demonstrated a clonal growth advantage, recipient mice appeared well with no evidence of overt hematologic disease for XX months” (if not). Along these lines, what were the peripheral blood counts and spleen sizes of the recipient mice shown in Figure 7 when they were euthanized? An elevated reticulocyte count can be a measure of ineffective erythropoiesis in mice (and humans) with MDS: were these measured or estimated from peripheral blood smears? Recticulocyte counts would have also been of interest for the secondary recipients shown in Figure 8 and Southern blot analysis to assess clonally at the level of vector integration would have also been informative. Finally, how did the disease phenotype compare in primary and secondary recipients with respect to the quantitative parameters outlined above?

---

## [Author Response]

*1) The reviewers are unconvinced by the argument that* MYBL2 *levels must be reduced to sub-haploinsufficient levels to cause myeloid malignancy. After all, Clarke et al show that deletion of a single copy of* MYBL2 *is sufficient to cause myeloproliferation in mice. The authors do not test an shRNA that only reduces* MYBL2 *levels by 50%, so they cannot rule out the possibility that* MYBL2 *could function as a typical haploinsufficient tumor suppressor. It would seem best to tone down this aspect of the Discussion*.

We agree with the reviewers and we have revised the Discussion to indicate that we cannot rule out that *MYBL2* could function as a haploinsufficient tumor suppressor (beginning “Clarke et al. have also described an association between *MBYL2* expression levels and the occurrence of hematologic disorders, including MDS, in aged *Mybl2* haploinsufficient mice”).

*2) The phenotypic description of the mice shown in Figure 7 is incomplete. First, it is unclear if hematologic disease was the cause of death in any of these mice (except for the one that developed AML) or if these findings were noted at the time of elective euthanasia. A Kaplan–Meier curve showing survival (if reduced) or a clear statement such as “although transduced cells expressing* Mybl2 *hairpins demonstrated a clonal growth advantage, recipient mice appeared well with no evidence of overt hematologic disease for XX months” (if not). Along these lines, what were the peripheral blood counts and spleen sizes of the recipient mice shown in Figure 7 when they were euthanized? An elevated reticulocyte count can be a measure of ineffective erythropoiesis in mice (and humans) with MDS: were these measured or estimated from peripheral blood smears? Recticulocyte counts would have also been of interest for the secondary recipients shown in Figure 8 and Southern blot analysis to assess clonally at the level of vector integration would have also been informative. Finally, how did the disease phenotype compare in primary and secondary recipients with respect to the quantitative parameters outlined above*?

We agree that this information is important and we have included new data in the figures, with new descriptions in the Results section to address these issues.

First, we extended our description of the phenotype by including the peripheral blood counts and spleen weights at the time of euthanasia in Figure 7. A new paragraph introducing these results has been added to the Results section (beginning “We also evaluated the circulating blood counts in 13 of the 17 mice at the time of euthanasia 6–7 months after transplant”).

Our strategy was to evaluate all of the mice at 6–7 months post-transplant, which is a time at which long-term hematopoietic reconstitution has been achieved in competitive repopulation assays. Thus, a Kaplan–Meier curve can’t be included as the mice were electively euthanized and the majority of the primary recipients appeared well at 6–7 months post-transplant.

The circulating blood abnormalities in the secondary animals appeared much more pronounced as illustrated in Figure 8, panel b. We now comment on this finding within the Results section, and the new paragraph begins “Analysis of blood samples collected from secondary recipients at 10 weeks revealed a marked clonal advantage for GFP+/Mybl2-RNAi cells transplanted from each of the four mice with primary disease”.